# Estimation of Cement Composites Fracture Parameters Using Deformation Criterion

**DOI:** 10.3390/ma12244206

**Published:** 2019-12-14

**Authors:** Marta Kosior-Kazberuk, Andrzej Kazberuk, Anna Bernatowicz

**Affiliations:** 1Faculty of Civil and Environmental Engineering, Bialystok University of Technology, 15-351 Bialystok, Poland; m.kosior@pb.edu.pl; 2Faculty of Mechanical Engineering, Bialystok University of Technology, 15-351 Bialystok, Poland; a.kazberuk@pb.edu.pl

**Keywords:** notch, deformation criterion, stress intensity factor, cement composite

## Abstract

A simple deformation criterion based on Dugdale’s cohesive zone model is presented. The criterion can be used for both the experimental determination of the critical stress intensity factor, *K_Ic_*, and the critical tip opening displacement, CTOD_c_. It can also be applied for the evaluation of the load capacity of structural elements. The criterion is presented in explicit and compact form, which allows straightforward calculations to be performed for the estimation of *K_Ic_* and CTOD_c_ values from the experimental data obtained from samples with a U-shaped notch, rounded with an arbitrary radius. Thanks to the simple form of the approximate relationship between the maximal load level and the dimensionless notch tip opening displacement, the reverse procedure was obtained, i.e., the estimation of the value of the maximal force loading the structural element as a function of the known critical stress intensity factor.

## 1. Introduction

The process of cracking an element of quasi-brittle material with a heterogeneous microstructure (e.g., cement composite) is associated with the formation and spreading of a major crack called a critical defect. The methods of fracture mechanics make it possible to describe the process of the destruction of such materials [1,2,3,4,5], as well as the degradation resulting from the impact of operational factors [6,7]. The parameters of material fracture mechanics, such as critical stress intensity factor, crack energy, and crack opening displacement, are determined experimentally using samples with cracks [8]. For metals, the procedure for determining these parameters has been standardized and is widely used. The cracking process begins with a fatigue-initiated crack in the notch tip.

In the case of cement composites and other brittle materials, the assessment of fracture mechanics parameters requires experimental tests using samples with a pre-formed primary crack or notch. Obtaining an initial crack of strictly defined geometric parameters is difficult. In standard quasi-brittle materials tests, an elongated U-shaped notch with a width of 1 to 10 mm is used, ignoring the effect of the rounding of its apex. Typically, a notch is produced at the stage of sample formation or by cutting with a diamond saw [9]. This approach is easy to implement, but rather narrow notches are obtained. Notches rounded with a greater radius are manufactured at the forming stage using reusable or disposable inserts. Notches of this kind are useful to analyze the influence of the notch geometry on the fracture process and especially to explain to what extent the geometry of the primary defect influences this. The crack propagation process should be analyzed based on the strength criteria of materials and on the basis of the criteria of fracture mechanics [10,11].

Most of the research on the influence of the shape of the notch on the fracture process has been focused on theoretical considerations rather than test methods and has been conducted using materials such as PMMA (poly(methyl methacrylate)) [10], metal alloys [11,12], and graphite [13]. The research results indicate a significant relationship between the notch rounding radius, the value of the damage force, and the cracking process evolution. Research conducted using concrete samples has mainly concerned the influence of the notch depth on fracture characteristics [6]. In the case of samples with notches where vertices are rounded with a large rounding radius (relative to the notch depth), the stress singularity disappears, and consequently, crack criteria based on linear elastic fracture mechanics cannot be used, even for brittle materials [12].

This work presents the application of a simple deformation criterion based on Dugdale’s cohesive model of the process zone [14], both for the experimental determination of the critical stress intensity factor and crack tip opening displacement as well as for the assessment of the loading capacity of the structural element.

## 2. Deformation Criterion

It is known that the limit equilibrium in the elastic–plastic body occurs when the opening displacement in the rounded notch tip is equal to its critical value:(1)$$\delta_{I}=\delta_{c}$$
where *δ_c_* is the critical value of the crack tip opening displacement (CTOD_c_) (for a sample in plane stress). It is assumed that *δ_c_* is constant for a particular material and that it depends on neither the notch opening angle (it is the same for a crack and for a V-shaped notch) nor the notch tip rounding radius, *ρ*.

Assuming small inelastic deformations, i.e., when the length of the inelastic band is small in comparison to the crack length and the size of the body (e.g., [15,16]), the opening displacement at the vertex of the notch is related to the critical stress intensity factor for a sample with a crack [17,18]. In plane stress conditions this relationship takes the form
(2)$$\delta_{c}=\frac{K_{c}^{2}}{E\sigma_{Y}}$$
where *E* is the modulus of elasticity and *σ_Y_* is the stress limit. Criterial Equation (1), for small inelastic deformations, can also be written in the following way [19]:(3)$${\tilde{\delta }}_{I}{\left(K_{I}\right)}^{2}=K_{c}^{2}$$
where ${\tilde{\delta }}_{I}$ is the dimensionless opening displacement at the tip of a rounded U-shaped notch and *K_I_* is the stress intensity factor at the sharp vertex of the corresponding crack.

Equation (3) can be used to determine the unknown failure load level, by which cracking starts, and the related stress intensity factor, *K_I_*, while the other values are regarded as known. Additionally, when the load that causes cracking is known and consequently the value of *K_I_* is known, Equation (3) can be used for the experimental determination of fracture toughness, *K_c_*, with the use of samples with rounded U-shaped notches.

The asymptotic solution to the elastic−plastic problem of an infinite rounded V-notch with a plastic band at the vertex was shown in [19,20]. The value of the dimensionless opening displacement at the notch vertex and the range of the inelastic zone are dependent on parameter *γ_Y_*, which is the dimensionless load level referenced to a material’s yield point [19,21], according to the following equation:(4)$$\gamma_{Y}=\frac{1}{\sqrt{2\pi \rho }}\frac{K_{I}}{\sigma_{Y}}.$$

The dimensionless values of the length of the inelastic zone, ${\tilde{l}}_{Y}$, arising from an infinite U-shaped notch and notch opening displacement, ${\tilde{\delta }}_{I}$, in function of parameter *γ_Y_* are shown in Figure 1 [19].

The graphs in Figure 1a,b, both for the length of the inelastic zone and the notch tip opening displacement, start from certain non-zero values of load level, because traction stress at the notch tip must achieve the yield point of the material:(5)$${\left(\sigma_{s}\right)}_{max}=\frac{K_{I}}{\sqrt{2\pi \rho }}R_{I}=\sigma_{Y}\to {\left(\gamma_{Y}\right)}_{min}=\frac{1}{R_{I}}=0.3341$$
where *R_I_* = 2.993 is the stress rounding factor [22,23]. For the U-shaped notch and parameter *γ_Y_* → ∞, the dimensionless inelastic band length approached ${\tilde{l}}_{Y}=\pi /8$ and the dimensionless notch opening displacement achieved ${\tilde{\delta }}_{I}=1$—the theoretical value of the crack opening displacement for a half-infinite crack with an inelastic band at the tip [24,25] (see also [26,27]).

The relationship between the dimensionless notch tip opening displacement, ${\tilde{\delta }}_{I}$, at the vertex of a U-shaped notch and the load level parameter, *γ_Y_*, can be approximated by
(6)$${\tilde{\delta }}_{I}={\left(1-a\gamma_{Y}^{b}\right)}^{2},\;\frac{1}{R_{I}}<\gamma_{Y}<\infty$$
where parameter *a* = 0.12124 was fitted by the least-squares method and the value of parameter *b*,
(7)$$b=\frac{\ln a}{\ln R_{I}}=-1.9247,$$
results from condition ${\tilde{\delta }}_{I}\left(1/R\right)=0$. The relative error of Equation (6) is less than 0.3% in the whole interval, $1/R_{I}<\gamma_{Y}<\infty$.

Allowing a less precise estimation of ${\tilde{\delta }}_{I}$ (with an error not exceeding 1%), the function ${\tilde{\delta }}_{I}\left(\gamma_{Y}\right)$ can be written down as
(8)$${\tilde{\delta }}_{I}={\left(1-{\left(R_{I}\gamma_{Y}\right)}^{-2}\right)}^{2},\;\frac{1}{R_{I}}<\gamma_{Y}<\infty .$$

The estimated relationship between the dimensionless crack length, ${\tilde{l}}_{Y}$, and parameter *γ_Y_* is described by the following equation:(9)$${\tilde{l}}_{Y}=\frac{\pi }{8}-c\gamma_{Y}^{d},\;\frac{1}{R_{I}}<\gamma_{Y}<\infty$$
where parameter *c* = 0.0088351 was fitted by the least-squares method and parameter *d*,
$$d=\frac{\ln\left(8a/\pi \right)}{\ln R_{I}}=-3.4611,$$
was calculated from the condition ${\tilde{l}}_{Y}\left(1/R\right)=0$. The relative error of the estimation of ${\tilde{l}}_{Y}$ from Equation (9) is less than 0.5% in a whole interval, $1/R_{I}<\gamma_{Y}<\infty$.

Assuming that a material’s parameters *K_c_* and *σ_Y_* and the geometrical characteristics of a sample are known, the fracture load can be determined by the equation
(10)$$K_{I}\sqrt{{\tilde{\delta }}_{I}}=K_{c}.$$

The approximate relationship in Equation (10) between the critical displacement in the notch tip and the dimensionless load level parameter, *γ_Y_*, can be written as
(11)$$\sqrt{\delta_{I}}=1-\frac{1}{R_{I}^{2}\gamma_{Y}^{2}}$$
where
(12)$$\gamma_{Y}=\frac{K_{I}}{\sigma_{Y}\sqrt{2\pi \rho }}$$

By linking Equations (11) and (12) to Equation (10) the following quadratic equation is obtained:(13)$$K_{I}\left[1-{\left(\frac{\sigma_{Y}\sqrt{2\pi \rho }}{R_{I}K_{I}}\right)}^{2}\right]=K_{c}\to K_{I}^{2}-K_{c}K_{I}=\frac{2\pi \rho \sigma_{Y}}{R_{I}^{2}}=0.$$

By inserting the values of the stress intensity factor, which depend on the shape of the sample and the loading conditions,
(14)$$K_{I}=F_{I}\sigma_{n}\sqrt{\pi l},$$
Equation (13) can be rewritten in the following form:(15)$$\sigma_{p}^{2}-\frac{K_{c}\sigma_{n}}{F_{I}\sqrt{\pi l}}-2\frac{\sigma_{Y}^{2}\varepsilon }{F_{I}^{2}R_{I}}=0.$$

Thus, the positive root of Equation (15) is equal to
(16)$$\frac{\sigma_{n}}{\sigma_{Y}}=\frac{1}{2F_{I}}\left[\sqrt{\frac{8\varepsilon }{R_{I}^{2}}+{\left(\frac{K_{c}}{\sigma_{Y}\sqrt{\pi l}}\right)}^{2}}+\frac{K_{c}}{\sigma_{Y}\sqrt{\pi l}}\right]$$
where ε=ρ/l is the relative notch radius at the vertex. For *ε* = 0 (sharp crack), the minimal value of nominal stress is obtained:(17)$${\frac{\sigma_{n}}{\sigma_{Y}}|}_{min}=\frac{1}{F_{I}}\left[\frac{K_{c}}{\sigma_{Y}\sqrt{\pi l}}\right].$$

## 3. Experimental Tests

For the experimental verification of the proposed deformation criterion, the results presented in [28], which analyzed the impact of the U-shaped notch rounding radius on the parameters of fracture mechanics calculated in accordance with the two-parameter Shah criterion [2,29] (recommended by RILEM [30] and similar to JCI standard [31]), were used. The most important elements of the description and selected results are listed below.

The samples were made of mortar based on Portland cement. A mixture of natural aggregate with a grain size not exceeding 2 mm was used. The reduction of grain size was aimed at eliminating the possible impact of coarse aggregate on cracking characteristics. The water−cement ratio (w/c) of the mortar was 0.35. The selected material made it possible to make custom test elements with different notch geometry. The average compressive strength of concrete after 28 days of curing was 51.7 MPa, the average tensile strength at bending was 4.01 MPa, and the tangential modulus of elasticity determined during the measurements of fracture mechanics parameters reached the value of 22.3 GPa. In beam samples with dimensions of 100 × 100 × 100 mm, notches 30 mm deep (ratio *a*_0_/*d* = 0.30) were produced during formation. The notch rounding radii were 0.15, 0.35, 0.65, 1.35, 3.15, 8.0, 15.0, and 30.0 mm. The tests were performed after 28 days of curing the samples in water at 20 ± 2 °C.

U-notched beam elements were loaded with a concentrated force under three-point bending conditions. The loading rate was chosen so that the maximum value was reached within about 5 min. The dimensions and the loading scheme of the sample element are shown in Figure 2.

During the tests, the diagram of the loading force, *P*, as a function of the notch mouth opening displacement was recorded. The design of the test stand provided the conditions for stabilized damage of the samples. For the measurements, a universal hydraulic testing machine MTS 322 (manufactured by MTS Systems Corporation in Eden Prairie in the USA) was used, enabling fast deformation control and load regulation. The results were recorded automatically. A view of the test stand with the sample element is shown in Figure 3.

## 4. Results Analysis

To determine the critical stress intensity factor using the deformation criterion, the only significant value is the maximum value of the force destroying the sample. The initial fragments of the P-CMOD relationship (shown in Figure 4 for all notch rounding radii) were numerically smoothed and piecewise approximated with the polynomial of third degree to remove measuring noise and obtain a repeatable determination of the maximum force value. 

The nominal stress value and the corresponding dimensionless load level factor were calculated for each sample:$$\sigma_{n}=\frac{3}{2}\frac{PS^{2}}{td^{2}},\gamma_{Y}=F_{I}\frac{\sigma_{n}}{\sigma_{Y}\sqrt{2\varepsilon }}$$
where the dimensionless stress intensity factor value is equal to *F_I_* = 1.042 (for *a*_0_/*d* = 0.3 [32]).

The dimensionless notch tip opening displacement corresponding to the value of *γ_Y_* was calculated according to Equation (6):$$\sqrt{{\tilde{\delta }}_{I}}=1-0.12124\gamma_{Y}^{-1.9247}.$$

The resulting values of the critical stress intensity factor were calculated using the relationship in Equation (3) in the form of
$$K_{c}=1.0426\sigma_{n}\sqrt{\delta_{I}\pi a_{0}}.$$

The obtained average values of the critical stress intensity factor for each analyzed relative notch rounding radius are shown in Figure 5. The value of *δ_c_* = CMOD_c_ was calculated from Equation (2).

By using the obtained average critical value of the stress intensity factor, *K_c_* = 0.46 MPa√m, the criterion values of nominal stresses destroying the tested element were calculated from Equation (16).

In dimensionless form, the criterion values are presented in Figure 6 with the experimental results.

## 5. Conclusions

The paper presents the application of the deformation criterion for determining the parameters of fracture mechanics of quasi-brittle materials using U-notched samples. Due to the ease of making this type of edge notch or hole with parallel edges and rounded vertices in samples at the forming stage, the criterion is particularly well suited for determining the parameters of concrete and cement composites.

This criterion has been presented in short form, which allows the simple calculation of the critical stress intensity factor, *K_c_*, and of the critical tip opening displacement, CTOD_c_, for any sample geometry with a rounded U-shaped notch and an arbitrary relative rounding radius. Due to the simple form of the approximating formulas used to calculate the dependence of the load level on the dimensionless notch opening displacement, formulas have also been obtained that enable the inverse procedure, i.e., the estimation of the destructive force of the structural element based on the known stress intensity factor.

## Figures and Tables

**Figure 1 materials-12-04206-f001:**
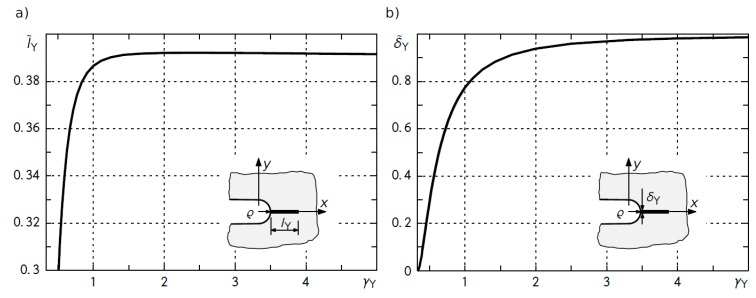
Relationship between the dimensionless length of an inelastic zone ${\tilde{l}}_{Y}$ (**a**) and the dimensionless opening displacement ${\tilde{\delta }}_{Y}$ (**b**) at the vertex of a U-notch and the dimensionless load level parameter *γ_Y_*.

**Figure 2 materials-12-04206-f002:**
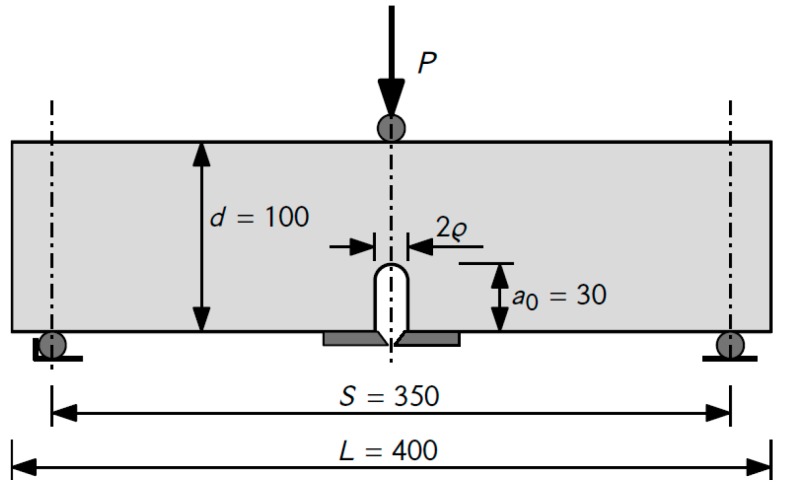
The loading scheme and dimensions of a sample with a U-shaped notch.

**Figure 3 materials-12-04206-f003:**
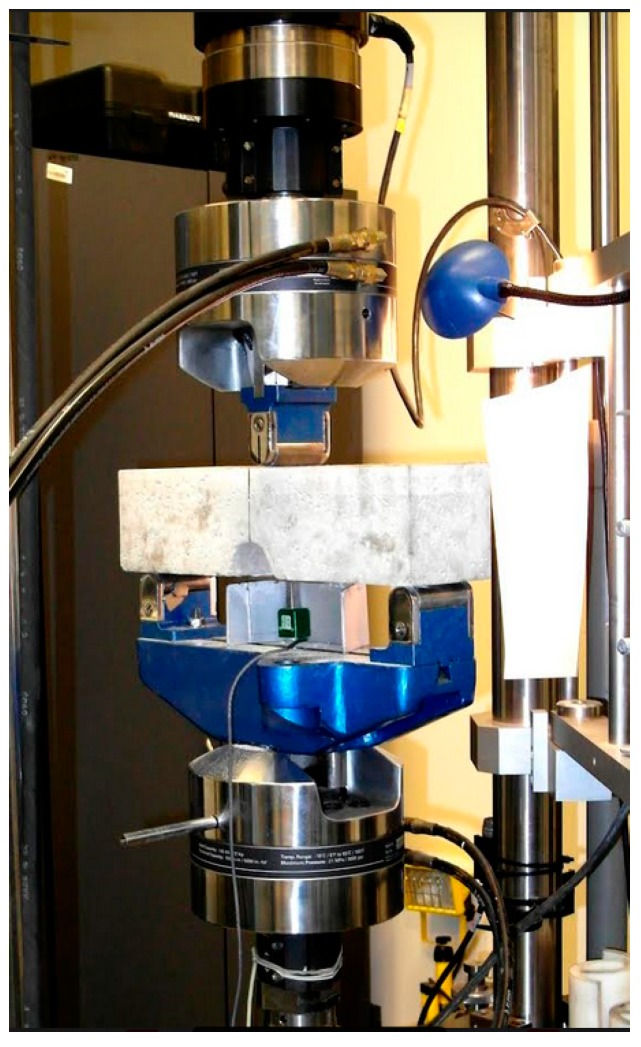
An image of the test stand.

**Figure 4 materials-12-04206-f004:**
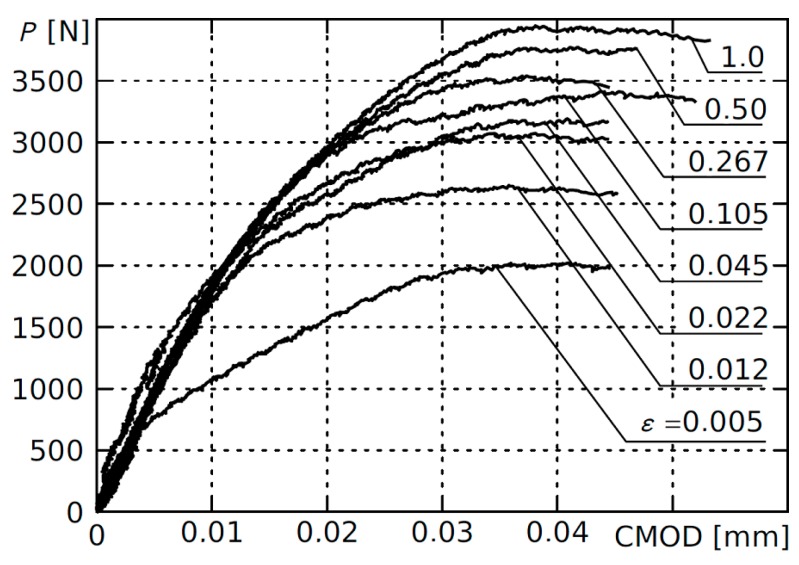
The initial phase of the P-CMOD graph for single-edge, U-notched samples with various relative notch rounding radii. *ε* = *ρ*/*a*_0_.

**Figure 5 materials-12-04206-f005:**
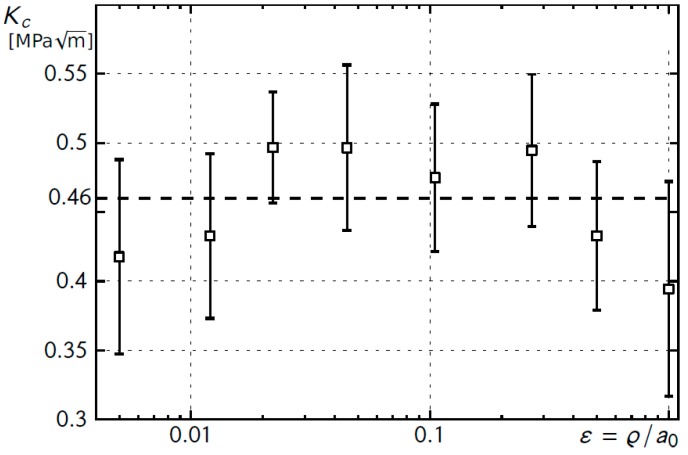
Average values of critical stress intensity factor, *K_c_*, as a function of relative U-notch rounding radius.

**Figure 6 materials-12-04206-f006:**
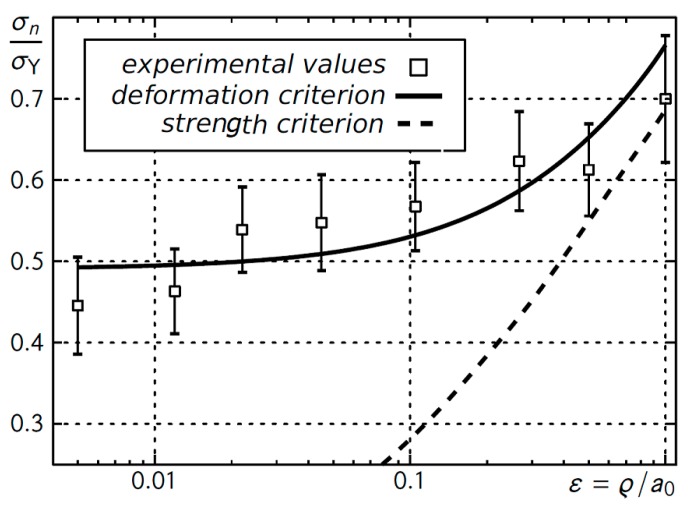
The critical values of nominal stresses for beams with U-shaped notches as a function of relative U-notch rounding radius.

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
