# Peer review of "Estimation of Cement Composites Fracture Parameters Using Deformation Criterion"

_materials, 2019, doi:10.3390/ma12244206_

Round 1

Reviewer 1 Report

The manuscript is devoted to the determination of fracture toughness in notched components using a defromation criterion based on Dugdale cohesive zone model. Some experimental tests on quasi-brittle specimens (mortar of Portlant cement) specimens are analysed.

The manuscript is interesting and relevant to the Journal. In my opinion it can be accepted for publication provided that the Authors carefully consider the following remarks:

Since the estimation criterion proposed is applied to a quasi-brittle material, it would appropriate to use also the term ‘inelastic’ instead of ‘plastic’ with reference to the fracture process zone. As a matter of fact, the former is a more appropriate expression for the actual mechanism of such a quasi-brittle material; The Authors refer to the two-parameter model of Shah and co-workers (e.g. it might be worth citing the recent paper on marble based on this model, https://doi.org/10.1111/ffe.12429). But also other methods dealing with fracture toughness estimation in quasi-brittle materials should be cited (e.g. see the RILEM and Japanese JCI-S-001 (2003) methods, and the 1990 effective crack model of Karihaloo and Nallathambi).

Author Response

Thank you very much for the suggestions. Of course, the term ‘inelastic’ is more appropriate to describe the behavior of FPZ in concrete element, so suggested corrections have been made. Also, we have added the following references:

RILEM TC QFS `Quasibrittle fracture scaling and size effect' - final report Materials and Structures, 2004, 37, 547-568 Karihaloo B. L., Nallathambi, P.: An improved effective crack model for the determination of fracture toughness of concrete, Cement and Concrete Research, 1989, 19, 603-610 Spagnoli A., Carpinteri A., Ferretti D., Vantadori S.: An experimental investigation on the quasi-brittle fracture of marble rocks, Fatigue & Fracture of Engineering Materials & Structures, 2016, 39, 956-968 SHAH S. CARPINTERI, A.: Fracture Mechanics Test Methods For Concrete, CRC Press, 2004 SHAH S.P.: Determination of fracture parameters (K Ic s and CTOD c) of plain concrete using three-point bend tests, Materials and Structures, Kluwer Academic Publishers, 1990, 23, 457-460 KITSUTAKA Y., KANAKUBO T.: Outline of JCI standard (JCI-S-001-003), Concrete Journal, 2006, 44, 10-15

Reviewer 2 Report

In this study, a deformation criterion is presented – on the basis of Dugdale’s cohesive zone model – for the evaluation of KIc and CTODc. This criterion is validated versus experimental investigations on U-shaped notch rounded samples. In addition, the method is applied to determine the ultimate load bearing capacity of the samples as a function of the KIc.

The paper is well-written. All required details are included in the text. The experimental procedure is clear and the findings are reasonable. Given the topic could be of interest to many readers of the journal, I would recommend its publication in the current form.

My only inquiry is in relation to the test procedure; why is the compressive strength of concrete after 90 days is used? I believe 28 days could work as well, while it saves huge amount of time.

Author Response

Thank you for a very favorable review. The ‘90-days’ curing time was placed by mistake.

Reviewer 3 Report

I suggest to publish the paper as it is, only with some spelling corrections. English corrections are welcome. I have found some typos at rows:

62: relationship takes

68: (other values are...)

73: gammaY, which is dimensionless...

75: values of length of 

188: factor.

Author Response

Thank you very much for the suggestions. All (I hope) typos have been corrected.
